# Reward-RAG: Enhancing RAG with Reward Driven Supervision

## Abstract

In this paper, we introduce Reward-RAG, a novel approach designed to enhance the Retrieval-Augmented Generation (RAG) model through Reward-Driven Supervision. Unlike previous RAG methodologies, which focus on training language models (LMs) to utilize external knowledge retrieved from external sources, our method adapts retrieval information to specific domains by employing CriticGPT to train a dedicated reward model. This reward model generates synthesized datasets for fine-tuning the RAG encoder, aligning its outputs more closely with human preferences. The versatility of our approach allows it to be effectively applied across various domains through domain-specific fine-tuning. We evaluate Reward-RAG on publicly available benchmarks from multiple domains, comparing it to state-of-the-art methods. Our experimental results demonstrate significant improvements in performance, highlighting the effectiveness of Reward-RAG in improving the relevance and quality of generated responses. These findings underscore the potential of integrating reward models with RAG to achieve superior outcomes in natural language generation tasks.

## 1 Introduction

Recent advancements in natural language processing have spurred the development of Retrieval-Augmented Generation (RAG) models, aimed at enhancing the quality and relevance of generated text by integrating external knowledge sources (Lewis et al., 2020; Guu et al., 2020; Izacard & Grave, 2021; Lin et al., 2024). These models leverage retrieved documents to provide contextually grounded responses, addressing inherent limitations in Large Language Models (LLMs) such as domain specificity (Siriwardhana et al., 2023; Xiong et al., 2024), and knowledge accuracy (Zhang et al., 2023; Kasai et al., 2023). In general, a retrieval system (Formal et al., 2022; Izacard et al., 2022; Wang et al., 2022a) first retrieves top-$k$ related documents for a question from an external database, then LLMs read the question and these documents to generate an answer.

The alignment between generated text and human preference remains a significant challenge for RAG approaches, particularly evident in question-answering tasks. Retrieval mechanisms often struggle to retrieve relevant information essential for specific queries (Zhang et al., 2024). State-of-the-art retrieval models can be categorized into dense retrieval and sparse retrieval (Luan et al., 2021). Sparse retrieval uses a sparse vector to represent statistical feature based on a vocabulary (Jones, 1972; Robertson & Zaragoza, 2009) which may fail to capture high level semantics, and suffer from the lexical gap (Berger et al., 2000; Izacard et al., 2022). On the other hand, dense retrieval leverages a pre-trained language model (PLM) to represent the input sequence by a fixed length vector (Reimers & Gurevych, 2019; Karpukhin et al., 2020) which may fail in specialized domains or with outdated data. Moreover, while PLMs excel in managing long-context windows(Su et al., 2024; Zhu et al., 2024; Ding et al., 2024), challenges arise with excessive retrieval context (Xu et al., 2024b; Liu et al., 2024a). Consequently, conventional retrieval pipelines typically adopt a two-stage process involving initial document retrieval followed by re-ranking (Chen et al., 2020; Glass et al., 2022; Ma et al., 2024). With these retrieval mechanisms, achieving a high recall rate is crucial for the success of a RAG system, and improving the system's ability to understand human preferences would indisputably elevate the relevance and quality of generated responses.

Based on the above discussions, we posit that achieving high recall with a concise list of pertinent context is crucial for developing RAG systems aligned with human preferences. Inspired by the suc-

cess of Reinforcement Learning from Human Feedback (RLHF) in aligning large language models (LLMs) with human preferences (Bai et al., 2022; Ouyang et al., 2022), we investigate its potential to adapt retrieval systems with a new reward model. Our proposed method, Reward-RAG, integrates reinforcement learning to augment RAG capabilities. Reward-RAG initiates by establishing reward models based on feedback indicating document relevance for specific queries. Since collecting human feedback is time-consuming and cost ineffective, we propose to utilize a CriticGPT to measure the relevance of retrieved documents and queries. CriticGPT is instructed to emulate human preferences using a small set of human preference examples. Leveraging these models, we fine-tune existing retrieval models within the RAG framework to retrieve high-quality content from external corpora. This approach aims to bridge the gap between general retrieval capabilities and the specific requirements of user preferences, thereby enhancing the relevance and quality of generated responses.

Our contribution can be summarized as follows:

- We propose Reward-RAG, a novel method that aligning RAG with human preferences by integrating a reward model into conventional RAG framework.
- We propose to utilize a CriticGPT in conjunction with human feedback which significantly reduce the amount of human preference data for training.
- We conduct experiments in different domains, compare our method with strong baselines in wide range RAG tasks as well as analyzing different aspects of our method to demonstrate the effectiveness including aligning RAG with new domains.

## 2 RELATED WORKS

**Large Language Models (LLMs)** has spurred significant advancements over the past few years. Beginning with GPT-1 (Radford et al., 2018) on the Transformer architecture (Vaswani et al., 2017), subsequent models like GPT-2 (Radford et al., 2019), GPT-3 (Brown et al., 2020), and the latest GPT-4 (OpenAI, 2024) have significantly enhance capabilities in text understanding and generation. Beyond the GPT series, models such as Mistral (Jiang et al., 2023), Gemini (Gemini Team, 2023), and LLaMA ((Touvron et al., 2023a), Touvron et al. (2023b)) demonstrate robust performance across various tasks like question-answering and entity recognition (Zhao et al., 2023). Training LLMs involves unsupervised pre-training, supervised fine-tuning, and alignment with human feedback, yet challenges persist in domain-specific tasks (Kandpal et al., 2023). Techniques like PEFT (Houlsby et al., 2019a) optimize fine-tuning efficiency, with emerging methods such as prompt-based learning (Lester et al., 2021; Li & Liang, 2021), adapters (Houlsby et al., 2019b; Fu et al., 2021; Wang et al., 2022c; He et al., 2022), and reparameterization (Hu et al., 2022; Edalati et al., 2022; Dettmers et al., 2023) showing promise by focusing on selective parameter adjustment for enhanced performance.

**Retrieval-Augmented Generation (RAG)** enhances LLM performance by expanding input with pertinent texts (Lewis et al., 2020; Guu et al., 2020). It integrates external database insights but faces key challenges: determining what, when, and how to retrieve documents (Gao et al., 2024). Khandelwal et al. (2020); Ram et al. (2023) study how to incorporate retrieval information into next token prediction pipeline. Guu et al. (2020); Borgeaud et al. (2022); Izacard et al. (2023); Zhang et al. (2024) propose an end-to-end training pipeline to fine-tuning existing LLMs to adapt with retrieval information. Chen et al. (2023a); Sarthi et al. (2024) analyze different types of knowledge representation in RAG. Methods like Dai et al. (2023) and Zhang et al. (2023) adjust retrieval models via contrastive learning and supervised fine-tuning, reliant on extensive datasets, posing scalability issues (Shi et al., 2024). Gutiérrez et al. (2024) introduces HippoRAG, a neurobiologically inspired retrieval system, by using knowledge graph to represent information as well as retrieve related passages. Combining RAG with RLHF is a promising direction, with Shinn et al. (2023) proposing episodic memory reinforcement, and (Kulkarni et al., 2024) proposing to train a policy agent to reduce the number of retrieval. Menick et al. (2022) leverage RLHF to train LLMs to generate answers with citing evidences from related documents for their claims. Asai et al. (2024) add special tokens to adaptively retrieve passages as well as generate and reflect on retrieved passages and its own generations, fine-tune their LLMs using an additional critic model. Zhou et al. (2023) refine models via reinforcement learning, but their reliance on LLMs' outputs complicates cost-efficiency. Our work focus on employing a reward model to enhance retrieval quality, specifically aiming to improve relevance and align with human preferences.

**Reinforcement Learning from Human Feedback (RLHF)** aligns LLMs with human values to mitigate biases and inaccuracies like hallucinations (Huang et al., 2023; Rauh et al., 2022). The first RLHF approach is RL-based, involving training reward models with preference datasets and fine-tuning policy models via algorithms like proximal policy optimization (Ouyang et al., 2022; Biderman et al., 2023; Schulman et al., 2017; Stiennon et al., 2020). The second method, Direct Preference Optimization (DPO), optimizes LLMs directly through supervised learning, sharing the RL-based approach's objective function (Rafailov et al., 2023; Morimura et al., 2024; Zeng et al., 2024). Reinforcement learning from AI feedback (RLAIF) is an attractive topic where LLMs are used to evaluate and guide the learning of other systems. Zheng et al. (2024) and Thomas et al. (2024) evaluate the alignment between AI feedback and human feedback in multiple scenarios. Our work introduces a novel approach using a reward model and CriticGPT to enhance retrieval-augmented generation.

## 3 METHODOLOGY

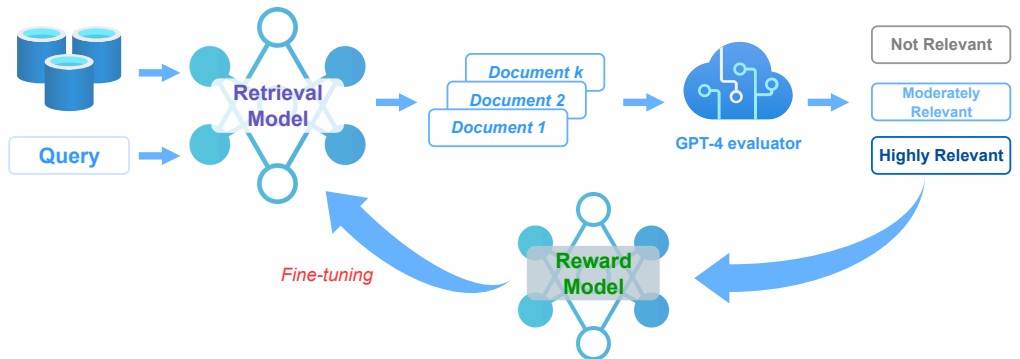

Figure 1: **Overview of our Reward-RAG**. Given a query and its knowledge database, a retrieval model is used to retrieve the top-$k$ relevant documents, which then are rated for the relevance by a CriticGPT. These $\langle query, document \rangle$ pairs and their CriticGPTs' feedback are used to train a reward model, which is used to fine-tune the RAG retrieval to better align with human preferences.

In this section, we present our Reward-RAG. We first describe the dense retrieval problem in RAG in section 3.1, then present how we apply reinforcement learning to this problem in section 3.2. Fig.2 illustrates the high-level design for Reward-RAG.

### 3.1 DENSE RETRIEVAL IN RAG

Let **Enc** denote the retrieval language model. Given a query $q$ and a document $d$, each with task-specific instructions $I_q$ and $I_d$, respectively, the embedding vectors are computed as follows: $e_q = \mathbf{Enc}(I_q \oplus q)$ and $e_d = \mathbf{Enc}(I_d \oplus d)$. The relevance score $sim(q, d)$ is determined by the cosine similarity between these two embedding vectors.

$$sim(q, d) = \frac{e_q . e_d}{\|e_q\| \|e_d\|} \quad (1)$$

In this work, we use both autoregressive and bidirectional language models (Devlin et al., 2019) as our retrieval models. We add two special tokens $[CLS]$ and $[EOS]$ to the list of tokens representing textual input:

$$[CLS], t_1, t_2, ..., t_n, [EOS] \quad (2)$$

where $t_1...t_n$ is the token representation of the input sequence. We use the embedding of the $[CLS]$ token and $[EOS]$ token from the last transformer layer as the vector representation of the input for the bidirectional language model and the autoregressive language model, respectively.

A crucial problem in RAG is how to retrieve relevant documents given a query (Gao et al., 2024), especially in domain-specific tasks where retrieval models can lack information compared to their

training data. We leverage a reward model to adapt the retrieval models for different tasks and user preferences effectively. The details of our approach will be introduced in the following sections.

## 3.2 USER PREFERENCE ALIGNMENT USING A REWARD MODEL

Inspired by RLHF, we design a mechanism to fine-tune the existing retrieval models to better align user preferences in the retrieved documents. We follow the RL-based design in RLHF, where we first build a reward model to evaluate the relevance between a query and a document, secondly we fine-tune retrieval models using the reward model (see Figure 2).

### 3.2.1 REWARD MODELS

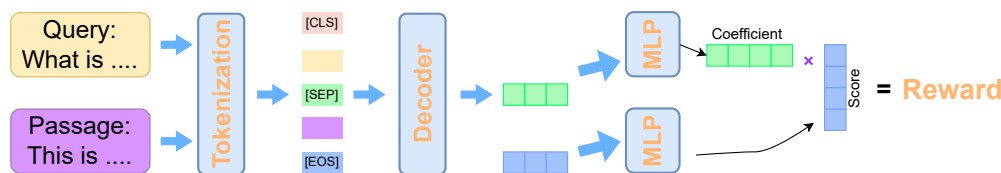

Figure 2: **Overview of the reward method**. We follow the design in Wang et al. (2024) to adapt an existing autoregressive model to be a reward model.

In RLHF, the reward model plays an important role in aligning LLMs, reflecting human values and expectations (Stiennon et al., 2020; Ouyang et al., 2022). We leverage GPT-4 as our CriticGPT to label the relevant level of a $\langle query, document \rangle$ pair as GPT-4 is proven to reach human-level accuracy for evaluation tasks (Liu et al., 2023; Hackl et al., 2023). The CriticGPT is instructed to mimic human preferences using a small set of human preference examples.

Our reward model is trained to rate the relevance between a question and a document corresponding to the feedback. This model involves providing the model with both a query and a candidate document as input, then produces a score representing the document's relevance to the query (Nogueira et al., 2019). Specifically, we construct the input from a query and a document as follow:

$$Input = [CLS], t_1^q, ..., t_{n_q}^q, [SEP], t_1^d, ..., t_{n_d}^d, [EOS] \tag{3}$$

where $[CLS]$ and $[EOS]$ are popular tokens in language processing to indicate the beginning and the ending of the input, $[SEP]$ is a special token to separate the query and the document, $t_1^q, ...t_{n_q}^q$ and $t_1^d, ...t_{n_d}^d$ are token sequences representing the query and the document respectively. We use Llama-3.1-8B-Instruct (Meta, 2024) as our pre-trained language model. We follow the design in Wang et al. (2024) to build the reward model. In more details, we first use the vector embedding of $[SEP]$ token and $[EOS]$ token from the final decoder layer as the vector representation for the query (denote as $Emb_q$) and the whole prompt input (denote as $Emb_p$) respectively. We then feed $Emb_p$ through a linear layer to obtain a $k$-dimensional vector prediction. To map the $k$-dimensional vector to a scalar reward, we calculate a coefficients vector by first using 2-layers MLP take the $Emb_q$ to output a $k$-dimensional vector, followed by a softmax function, then multiply the coefficients vector to the reward vector prediction from $Emb_p$.

$$Emb_q = Decoder(Input)[-1][SEP]$$
$$Emb_p = Decoder(Input)[-1][EOS]$$
$$V_{reward} = Linear(Emb_p)$$
$$Coeff = \text{softmax}(MLP(Emb_q))$$
$$r_\theta(q, d) = Coeff^T * V_{reward}$$

We use mean square error as our loss function to train the reward model:

$$loss(\theta) = E_{(\langle q, d \rangle, w) \sim D}[(r_\theta(q, d) - w)^2] \tag{4}$$

where $r_\theta(q, d)$ is the scalar reward for a query $q$ and a document $d$ from the reward model parameterized by $\theta$, $w$ is the expected reward, and $D$ is the feedback dataset.

### 3.2.2 COLLECTING LLMS' FEEDBACK

Relevance assessments by human annotators are time-consuming, labor-intensive, and costly. In our works, we use LLMs to judge the relevancy between a query and a passage or a document. There are two main problems in this phase:

- *Sampling*: For a query, how to sample documents from a corpus to evaluate the relevance?
- *Prompting*: How can we teach the LLMs by prompts to align with human assessments?

de Souza P. Moreira et al. (2024) studies hard-negative mining methods in fine-tuning retrieval embeddings. Following their works, we first use an existing retrieval encoder from the MTEB leaderboard (Muennighoff et al., 2023) to retrieve the top-25 related documents for each query, we then pick the top document and sample another 4 documents after ignoring documents have relevance scores higher than a threshold calculated from the highest score.

Writing a good prompt to align LLMs' output with human preferences is another crucial issue. Following the analysis in (Zheng et al., 2024; Thomas et al., 2024), we instruct LLMs by decomposing the problem into step-by-step tasks. The details of prompts and our analysis of different prompts is in Appendix B. After collecting LLMs' feedback for selected $\langle query, document \rangle$ pairs, we train the reward model and use the reward model to rate the top-25 related documents for a query.

### 3.3 FINE-TUNING RETRIEVAL MODEL

Given the reward model, we first synthesize $\langle query, document, reward \rangle$ data to fine-tune the retrieval model. We perform hard-negative mining by firstly using a retrieval model to retrieve top-50 related documents for a query, followed by rating the relevance for each pair using the reward model. We use a threshold to determine which $\langle query, document \rangle$ pairs are positive sample and use the rest as hard-negative samples.

We use InfoNCE loss (van den Oord et al., 2019) as the objective function to fine-tune retrieval models. Given a query $q$, a positive document $d^+$, and a set of negative documents $D^-$, the loss function is represented as:

$$\mathcal{L}(q, d^+, D^-) = -\log \frac{\exp(sim(q, d^+))}{\exp(sim(q, d^+)) + \sum\limits_{d^- \in D^-} \exp(sim(q, d^-))} \quad (5)$$

where $sim(q, d)$ is the similarity value between a query $d$ and a document $d$ defined in equation (1). For efficient training, the negative set $D^-$ includes both hard-negative and in-batch negatives, which are derived from positive documents and hard negative documents associated with other queries. This training pipeline tends to benefit from a bigger set of negative samples. During the inference phase, we keep the same pipeline as in a typical RAG system. In more details, we first embed the external database using the fine-tuned retrieval model, then perform retrieving with a fast $k$-nearest neighbors library such as FAISS (Johnson et al., 2021).

## 4 EXPERIMENTS

In this section, we present our experiments in a wide range of NLP tasks as well as analyzing our models in different aspects.

### 4.1 MAIN EXPERIMENTS

#### 4.1.1 EXPERIMENTS SETUP

**Tasks and Datasets.** We first conduct experiments on general domains: (1) *Open-domain QA*, which includes Natural Questions (NQ) (Kwiatkowski et al., 2019), and TriviaQA (Joshi et al., 2017). (2)

Table 1: Performance of our encoder and comparison with existing state-of-the-art models at the same size. NDCG@10 is used as the metric to benchmark retrieval encoders. These models are benchmark on three datasets from MTEB Benchmark (Muennighoff et al., 2023).

| Task | NQ | HotPotQA | Fever |
|---|---|---|---|
| SPLADE++ (Formal et al., 2022) | 54.4 | 68.6 | 79.6 |
| Promptgator (Dai et al., 2023) | - | 60.4 | 76.2 |
| Contriever (Izacard et al., 2022) | 49.5 | 63.8 | 75.8 |
| Dragon (Lin et al., 2023) | 53.7 | 66.2 | 78.1 |
| Gte-large-v1.5 Li et al. (2023) | 56.8 | 68.2 | **93.8** |
| Bge-large-v1.5 Xiao et al. (2024) | 55.0 | **74.1** | 87.2 |
| E5-large-unsupervised (Wang et al., 2022b) | 41.7 | 52.2 | 68.6 |
| UAE-large-v1 (Li & Li, 2024) | 55.8 | 73.1 | 88.2 |
| E5-large-unsupervised (ours) | **60.0** | 65.4 | 76.3 |

*Fact verification* includes FEVER (Thorne et al., 2018). We use the split from the KILT benchmark (Petroni et al., 2021).

**Training data and settings.** We use Natural Questions (NQ) (Kwiatkowski et al., 2019), Trivia-QA (Tri) (Joshi et al., 2017), and SQUAD (Rajpurkar et al., 2016) to build our models for general domain question-answering tasks. We follow the design in DPR (Karpukhin et al., 2020) to use preprocessed 2018 English Wikipedia as our corpus. To train the reward model, we sample 9000 queries from the NQ dataset and 3-5 documents for each query to label the relevance. For fine-tuning the retrieval encoder, we use a total of 100k queries from a blend of NQ and TriviaQA datasets as our train set and use our reward models to mine positive and negative documents as explained above.

**Baselines.** We consider baselines in terms of text retrieval and question-answering tasks. For text retrieval, we consider Promptgator (Dai et al., 2023), Dragon (Lin et al., 2023), Contriever (Izacard et al., 2022), SPLADE++ (Formal et al., 2022), GTE Li et al. (2023). For question-answering, we consider baseline LLMs without RAG (Mixtral-8x22B-Instruct (Jiang et al., 2023), PaLM2 (Anil et al., 2023), GPT-3.5-turbo (OpenAI, 2022), GPT-4 (OpenAI, 2024)), baselines with retrieval (Atlas (Izacard et al., 2023), Raven (Huang et al., 2024), Self-RAG (Asai et al., 2024), Recomp (Xu et al., 2024a), Replug (Shi et al., 2024), Ra-dit (Lin et al., 2024), ChatQA-1.5 (Liu et al., 2024b), RankRAG (Yu et al., 2024), and RAG pipeline using LLMs)

**Evaluation Metrics.** For Open-domain QA tasks, we use *Exact Match (EM)* as the main metric for NQ and TriviaQA. We also report *accuracy* for TriviaQA. For *Fact verification* task, we use *accuracy* as the main metric.

**Implementation Details.** We use *E5-large-unsupervised* (Wang et al., 2022b) as our base retrieval encoder to fine-tune. We use the baseline encoder to retrieve top-25 documents from the Wikipedia corpus for each query in our training set and sample from these documents to rate the relevance of $\langle query, document \rangle$ pairs using GPT-4o. We use Llama-3.1-8B-Instruct (Meta, 2024) as our critic model. We apply LoRA (Hu et al., 2022) and DeepSpeed (Rasley et al., 2020) to train our models efficiently. The detailed training settings and prompts is in Appendix A.

### 4.1.2 RESULTS

We first measure our retrieval encoders in the information retrieval task. We use three datasets in the general domain from the MTEB benchmark (Muennighoff et al., 2023) to test our model. We report the NDCG@10 score of our models and compare them with baselines and state-of-the-art models. Table 1 represents the performance of our model and another baseline. As our model has less than 400M parameters, we only select state-of-the-art models that have similar number of parameters from the MTEB leaderboard. Compared to the base model, our models increase performance on both three datasets. On the NQ dataset, our model is the best model.

Results for the downstream question-answering tasks are shown in Table 2. On the NQ and FEVER datasets, our model archives the best performance, while on the TriviaQA dataset, our method is the second-best model. It is noteworthy that in other models including RA-DIT, RankRAG, and Self-RAG, their methods fine-tune LLMs to adapt to downstream tasks, which is expensive and limits the

Table 2: Results of Reward-RAG and baselines in general domains on differernt datasets. We use the split from KILT benchmark for our results. Results unavailable in public reports are marked as "–"

| Task | NQ | TriviaQA | FEVER |
|---|---|---|---|
| Metric | EM | EM / Acc | Acc |
| Without Retrieval-Augmented Generation | | | |
| PaLM2 540B (Anil et al., 2023) | 37.1 | 86.1/- | - |
| Mixtral-8x22B-Instruct (Jiang et al., 2023) | 40.1 | 82.2/- | - |
| GPT-3.5-turbo-1106 (OpenAI, 2022) | 38.6 | 82.9/91.7 | 82.7 |
| GPT-4-0613 (OpenAI, 2024) | 40.3 | **84.8**/94.5 | 87.7 |
| With Retrieval-Augmented Generation | | | |
| Atlas 11B (Izacard et al., 2023) | 26.7 | 56.9/- | 77.0 |
| Raven 11B (Huang et al., 2024) | 29.6 | 65.7/- | - |
| Self-RAG 7B (Asai et al., 2024) | - | -/66.4 | - |
| Self-RAG 13B (Asai et al., 2024) | - | -/69.3 | - |
| RECOMP 20B (Xu et al., 2024a) | 37.0 | 59.0/- | - |
| RePlug 65B (Shi et al., 2024) | 28.8 | 72.6/- | 73.3 |
| RA-DIT 65B (Lin et al., 2024) | 35.2 | 75.4/- | 80.7 |
| Llama3-ChatQA-1.5 8B (Liu et al., 2024b) | 42.4 | 81.0/87.6 | 90.9 |
| Llama3-RankRAG 8B (Yu et al., 2024) | 50.6 | 82.9/89.5 | 92.0 |
| GPT-3.5-turbo-1106 RAG (ours) | 42.2 | 75.6/80.4 | 89.8 |
| GPT-4-0613 RAG (ours) | **50.9** | 84.4/90.5 | **92.3** |

generalization of LLMs. In our method, we do not modify the LLMs; instead, we aim to guide them by providing valuable information in a cost-effective way. In Table 5, we show a sample query from the NQ dataset with retrieved documents and the answer from different models. We observe that when the correct answer appears multiple times in the provided contexts, the presence of distractors does not affect the LLMs' responses.

## 4.2 DOMAIN SPECIFIC RAG TASKS

**Tasks and Datasets.** Besides general domain, we study the performance of our method in the medical field. We use Mirage (Xiong et al., 2024), a recent RAG benchmark, to test our method. There are 5 dataset in their benchmark: PubMedQA (Jin et al., 2019), BioASQ (Tsatsaronis et al., 2015)), MMLU-med (Hendrycks et al., 2021), MedMCQA (Pal et al., 2022), MedQA (Jin et al., 2021). Followed (Xiong et al., 2024), we use MedCorp[1] as our corpus.

**Results.** Table 4.2 shows the performance of our models and other baselines. We report the accuracy as the format of the downstream task is multiple-choice questions. Our method outperforms other baselines on the PubmedQA dataset, while it is the second-best model on the BioASQ dataset. Table 6 shows case studies we pick from different datasets. Since questions in the medical domain require logical thinking and reasoning, we emphasize the importance of providing correct relevant documents.

## 4.3 ABLATION STUDIES

### 4.3.1 COMPARE FEEDBACK FROM DIFFERENT LLMS

In order to compare the feedback collected from different LLMs, we calculate the confusion matrix between them on a subset of our dataset. We use the same prompt to collect feedback from GPT-3.5 and GPT-4o. Figure 3 shows the confusion matrix of the two models' feedback. In total, the percentage of agreement is $61.3\%$, presenting a huge gap between these two models. We sampled 50 queries along with their corresponding documents to evaluate the quality of feedback from these two models. The qualitative results indicate that the feedback from GPT-4o is better and more consistent than that from GPT-3.5. Therefore, we use GPT-4o to label data for our experiments.

---

[1]https://huggingface.co/MedRAG

Table 3: Results of Reward-RAG and baselines in the medical field on Mirage benchmark. For baseline using retrieval-augmented generation, MedCorp is used as corpus and RRF-4 is the retrieval method, most numbers are from public reports (Xiong et al., 2024; Yu et al., 2024)

| Task | MMLU-med | PubmedQA | BioASQ | MedQA | MedMCQA |
|---|---|---|---|---|---|
| Without Retrieval-Augmented Generation | | | | | |
| GPT-3.5 (OpenAI, 2022) | 72.9 | 36.0 | 74.3 | 65.0 | 55.2 |
| GPT-4-0613 (OpenAI, 2024) | **89.4** | 39.6 | 84.3 | **83.9** | **69.8** |
| PMC-llama 13B (Wu et al., 2024) | 52.2 | 55.8 | 63.1 | 44.4 | 46.6 |
| Llama2 70B (Touvron et al., 2023b) | 57.4 | 42.2 | 61.2 | 47.8 | 42.6 |
| Mixtral 8*7B (Jiang et al., 2024) | 74.0 | 35.2 | 77.5 | 64.1 | 56.2 |
| Meditron 70B (Chen et al., 2023b) | 64.9 | 53.4 | 68.4 | 51.6 | 46.7 |
| With Retrieval-Augmented Generation | | | | | |
| GPT-3.5 (OpenAI, 2022) | 75.5 | 67.4 | 90.3 | 66.6 | 58.0 |
| GPT-4-0613 (OpenAI, 2024) | 87.2 | 70.6 | **92.6** | 82.8 | 66.6 |
| PMC-llama 13B (Wu et al., 2024) | 52.5 | 42.6 | 48.3 | 56.0 | 65.2 |
| Llama2 70B (Touvron et al., 2023b) | 54.5 | 50.4 | 73.9 | 44.9 | 43.1 |
| Mixtral 8*7B (Jiang et al., 2024) | 75.8 | 67.6 | 87.5 | 60.0 | 56.4 |
| Meditron 70B (Chen et al., 2023b) | 65.4 | 56.4 | 76.8 | 49.5 | 52.6 |
| Llama3-ChatQA-1.5 8B (Liu et al., 2024b) | 61.4 | 66.4 | 82.7 | 42.4 | 46.9 |
| Llama3-RankRAG 8B (Yu et al., 2024) | 64.5 | 65.0 | 84.4 | 48.8 | 56.9 |
| GPT-3.5-turbo-1106 RAG (ours) | 69.7 | 69.2 | 89.5 | 59.2 | 52.4 |
| GPT-4-0613 RAG (ours) | 84.4 | **70.8** | 90.3 | 64.5 | 57.4 |

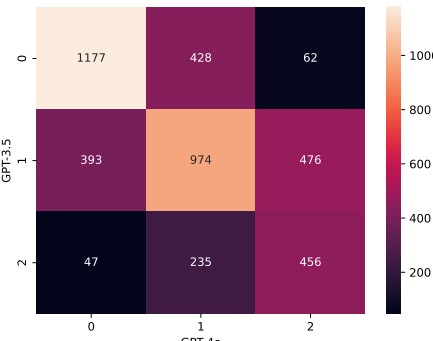

Figure 3: Confusion matrix between GPT-3.5's feedback and GPT-4o's feedback. Labels 0, 1, and 2 are corresponding to *Not Relevant*, *Moderately Relevant*, and *Highly Relevant* respectively.

### 4.3.2 PROMPTS FOR FEEDBACK COLLECTION

As we use black box LLMs to annotate data, prompts are the only way to control their quality. In our work, we try different prompting techniques including in-context learning and the design by Thomas et al. (2024). Specifically, for in-context learning, we provide ten $\langle query, document \rangle$ pairs to teach LLMs how to rate. For the other method, instead of providing examples, we split the task into sub-tasks that can be easier to answer and we ask the model to answer these questions before rating the relevance between a query and a document. Inspired by Wei et al. (2022), we call this method *"think step-by-step"*. The details of these two prompts are shown in Appendix B.

We qualitatively assess the annotations of GPT-4o using different prompts. We sampled 50 queries to evaluate how many responses from GPT-4o were incorrect compared to the ground truth answers. For the in-context learning, the accuracy is 0.7. We found that the most common types of errors are hallucinations and implications, particularly when the passage mentions the ground truth answer but in a context that is different from the query. For the *"think step-by-step"* prompt, the accuracy is 0.83. This is because LLMs answer a series of questions before making a final decision, which results in more consistent and robust annotations.

### 4.3.3 CASE STUDY

In Table 4 we present a case study from the NQ dataset where the human annotation for positive documents is incorrect, while the annotation by our reward model is accurate. For this query, the correct answer (The White Rabbit) is mentioned in both passages but in the document labeled by

Table 4: A case study on the positive document picked by the reward model compare to human annotations

| Query | who said i 'm late i 'm late for a very important date |
|---|---|
| Human labeled | White Rabbit The White Rabbit is a fictional character in Lewis Carroll's book ""Alice's Adventures in Wonderland"". He appears at the very beginning of the book, in chapter one, wearing a waistcoat, and muttering ""Oh dear! Oh dear! I shall be too late!"" Alice follows him down the rabbit hole into Wonderland. Alice encounters him again when he mistakes her for his housemaid Mary Ann and she becomes trapped in his house after growing too large. The Rabbit shows up again in the last few chapters, as a herald-like servant of the King and Queen of Hearts. In his article 
 Note: Incorrect label (the passage does not related to the query) |
| Reward model | the ""monster"" (Alice) out of his house, Dodo's ultimate solution is to burn the house down, to which the White Rabbit is greatly opposed. At the Mad Tea Party, the Mad Hatter and the March Hare try to ""fix"" his watch, proclaiming it ""exactly two days slow"". Through various food they put in the watch (butter, tea, jam, and lemon), the two cause it to go mad, and the Hare smashes it with his mallet. The Rabbit was perhaps most famous for the little ditty he sang at the beginning, ""I'm late! I'm late! For a very important date! No 
 Note: Correct label |

Table 5: A case study on the top-retrieved context and predictions on NQ dataset. We use the example query from RankRAG(Yu et al., 2024) to compare their models with ours. Red text denotes distractors, while green stands for evidences.

| | **Q**: who hosted and won the inagural world cup? **A**: Uruguay |
|---|---|
| ChatQA-1.5 | **Passage 1**: FIFA World Cup second round on home soil in 1982. England (1966) won its only title while playing as a host nation. Uruguay (1930), Italy (1934), Argentina (1978) and France (1998) won their first titles as host nations but have gone on to win again, while Germany (1974) won their second title on home soil... 
 **Passage 2**: FIFA World Cup hosts country is now chosen in a vote by FIFA's Congress ... Only Mexico, Italy, France, Germany (West Germany) until shortly after the 1990 World Cup) and Brazil have hosted the event on two occasions. 
 **Passage 3**: CONCACAF hosts, beating the bids of Canada and the United States, and thereby became the first nation to host two World Cups. This second World Cup in Mexico came 16 years after the first one in 1970... 
 **Passage 4**: 1998 FIFA World Cup Africa made their first appearances in the finals. France was awarded the 1998 World Cup on 2 July 1992 by the executive committee of FIFA during a general meeting in Zürich, Switzerland. They defeated Morocco by 12 votes to 7. 
 **Passage 5**: 2026 FIFA World Cup be hosted by one of the remaining four confederations: CONCACAF (North America; last hosted in 1994), CAF (Africa; last hosted in 2010), CONMEBOL (South America; last hosted in 2014), or OFC (Oceania, never hosted before)...    **Prediction**: *Germany* (✗) |
| RankRAG | **Passage 1**: FIFA World Cup second round on home soil in 1982. England (1966) won its only title while playing as a host nation. Uruguay (1930), Italy (1934), Argentina (1978) and France (1998) won their first titles as host nations but have gone on to win again, while Germany (1974) won their second title on home soil... 
 **Passage 2**: Timeline of association football penalty kicks. Thirteen teams enter the first World Cup, held in Uruguay. The hosts beat Argentina 4–2 in the final. Contested between the top national teams of continental Europe, Dr. Gerö Cup' first edition is won by Italy. 
 **Passage 3**: The Uruguay national football team represents Uruguay in international association football and is controlled by the Uruguayan Football Association. They have won the Copa América 15 times, the most successful national team in the tournament, the most recent title being the 2011 edition. The team has won the FIFA World Cup twice, including the first World Cup in 1930 as hosts, defeating Argentina 4–2 in the final. 
 **Passage 4**: FIFA World Cup hosts country is now chosen in a vote by FIFA's Congress. The decision is currently made roughly seven years in advance of the tournament, though the hosts for the 2022 tournament were chosen at the same time as those for the 2018 tournament. 
 **Passage 5**: CONCACAF hosts, beating the bids of Canada and the United States, and thereby became the first nation to host two World Cups. This second World Cup in Mexico came 16 years after the first one in 1970...    **Prediction**: *Uruguay* (✓) |
| Reward-RAG | **Passage 1**: The first two World Cup matches took place simultaneously on 13 July 1930, and were won by France and the USA, who defeated Mexico 4–1 and Belgium 3–0 respectively. The first goal in World Cup history was scored by Lucien Laurent of France. In the final, Uruguay defeated Argentina 4–2 in front of 93,000 people in Montevideo, and became the first nation to win the World Cup. After the creation of the World Cup, 
 **Passage 2**: 1950 FIFA World Cup The 1950 FIFA World Cup, held in Brazil from 24 June to 16 July 1950, was the fourth FIFA World Cup. It was the first World Cup since 1938, the planned 1942 and 1946 competitions having been cancelled due to World War II. It was won by Uruguay, who had won the inaugural competition in 1930... 
 **Passage 3**: but the choice of Uruguay as a venue for the competition meant a long and costly trip across the Atlantic Ocean for European sides. No European country pledged to send a team until two months before the start of the competition. 
 **Passage 4**: The 1978 FIFA World Cup, the 11th staging of the FIFA World Cup, quadrennial international football world championship tournament, was held in Argentina between 1 and 25 June. The Cup was won by the Argentine hosts, who defeated the Netherlands 3–1 in the final... 
 **Passage 5**: the first World Cup coincided with the centennial anniversary of the first Constitution of Uruguay. For that reason, the main stadium built in Montevideo for the World Cup was named Estadio Centenario.    **Prediction**: *Uruguay* (✓) |

human, it is not related to the query, on the other hand, the passage labeled by reward model answers the query with a clear evidence. More samples are provided in the appendix.

## 5 CONCLUSION

In conclusion, our study highlights the transformative potential of integrating synthetic data with a dedicated reward model in enhancing the performance of Retrieval-Augmented Generation (RAG) systems. By utilizing CriticGPT to generate tailored datasets, we enable general-domains and specific-domains fine-tuning that aligns model outputs more closely with human preferences. This synergy not only improves the relevance and quality of generated responses but also demonstrates advancements over existing state-of-the-art methods. The promising results from our evaluations across various domains affirm that the combination of synthetic data and reward-driven supervision can elevate the capabilities of RAG, paving the way for more effective natural language generation applications.

Table 6: A case study on the medical domain. We select sample questions from different datasets, demonstrate the retrieved documents with LLM's answer of our method. Green stands for evidences.

| | |
|---|---|
| **MedMCQA** | **Query**: The clot formed is not stable unless extensive cross-linking occurs. This extensive cross-linking of blood clot is done by:
**Choices**: A: "Plasmin", B: "Thrombin", C: "HMWK", D: "Factor XIII"
**Documents**: Covalent cross-linking of fibrin chains is required for stable blood clot formation, which is catalyzed by coagulation factor XIII (FXIII), a proenzyme of plasma transglutaminase consisting of catalytic A (FXIII-A) and non-catalytic B subunits (FXIII-B). Herein, we demonstrate that FXIII-B accelerates fibrin cross-linking. Depletion of FXIII-B from normal plasma supplemented with a physiological level of recombinant FXIII-A resulted in delayed fibrin cross-linking, reduced incorporation of FXIII-A into fibrin clots, and impaired activation peptide cleavage by thrombin; the addition of recombinant FXIII-B restored normal fibrin cross-linking, FXIII-A incorporation into fibrin clots, and activation peptide cleavage by thrombin. Immunoprecipitation with an anti-fibrinogen antibody revealed an interaction between the FXIII heterotetramer and fibrinogen mediated by FXIII-B and not FXIII-A.
**LLMs' output**: Document 2 mentions that covalent cross-linking of fibrin chains is required for stable blood clot formation and that factor XIII (FXIII) mediates clot stability by covalent cross-linking of fibrin-fibrils and inhibition of fibrinolysis. Therefore, the extensive cross-linking of blood clot is done by Factor XIII (B).       **Prediction**: B (✓) |
| **BioASQ** | **Query**: Is Tilavonemab effective for progressive supranuclear palsy?
**Choices**: A: "yes", B: "no"
**Document**: Progressive supranuclear palsy is a neurodegenerative disorder associated with tau protein aggregation. Tilavonemab (ABBV-8E12) is a monoclonal antibody that binds to the N-terminus of human tau. We assessed the safety and efficacy of tilavonemab for the treatment of progressive supranuclear palsy. We did a phase 2, multicentre, randomised, placebo-controlled, double-blind study in Australia, Canada, France, Germany, Italy, Japan, Spain, and the USA....Between Dec 12, 2016, and Dec 31, 2018, 466 participants were screened, 378 were randomised. The study was terminated on July 3, 2019, after prespecified futility criteria were met at the second interim analysis. A total of 377 participants received at least one dose of study drug and were included in the efficacy and safety analyses (2000 mg, n=126; 4000 mg, n=125; placebo, n=126)... A similar safety profile was seen in all treatment groups. No beneficial treatment effects were recorded. Although this study did not provide evidence of efficacy in progressive supranuclear palsy, the findings provide potentially useful information for future investigations of passive immunisation using tau antibodies for progressive supranuclear palsy.
**LLMs' output**: Based on the provided documents, a phase 2, multicentre, randomised, placebo-controlled, double-blind study was conducted to assess the safety and efficacy of Tilavonemab for the treatment of progressive supranuclear palsy. The study involved participants diagnosed with possible or probable progressive supranuclear palsy who were symptomatic for less than 5 years. The primary endpoint was the change from baseline to week 52 in the Progressive Supranuclear Palsy Rating Scale (PSPRS) total score. The study was terminated after prespecified futility criteria were met at the second interim analysis. The findings showed that Tilavonemab did not provide evidence of efficacy in progressive supranuclear palsy.
**Prediction**: B (✓) |
| **MMLU** | **Query**: When an influenza virus enters a cell, it immediately starts to do which of the following?"
**Choices**: A: "Incorporate viral DNA into the host cell's chromosome", B: "Destroy the host cell's transcriptional machinery", C: "Replicate its genetic material and synthesize viral proteins", D: "Use a viral copy of reverse transcriptase to manufacture viral DNA"
**Document 1**: Inhibitors of glycolysis, oxidative phosphorylation, protein synthesis, membrane Na+-K+ transport and microfilament and microtubule function have been employed to elucidate the mechanism of influenza virus uptake by CAM and CEF cells. Electron microscopy demonstrated uptake of virus by viropexis in the presence of all these inhibitors. Utilizing a pulse labelling technique, virus entering CEF cells in the presence of inhibitors was shown to initiate specific virus polypeptide synthesis after neutralization of remaining extracellular virus and removal of the inhibitors. As a consequence of these findings an energy independent mechanism of viropexis has been proposed.
**Document 2**: As a major pathogen of human and certain animal species, influenza virus causes wide spread and potentially devastating disease. To initiate infection, the virus first binds to cellular receptors comprising either -(2,3 ) or -(2,6) linked sialic acid. Recent advances in our understanding of the influenza virus receptor and viral host species involved have shed light on the molecular mechanism of how influenza virus transmits across species and adapts to a new host. Following receptor binding, influenza viruses are internalized through multiple endocytic pathways, including both clathrin- and non-clathrin-dependent routes, which have recently been visualized at single viral particle level. The viral envelope then fuses with the endosomal membrane in a low pH-dependent manner and the viral genome is released into the cytosol, followed by further transport to the nucleus where genome replication occurs.
**LLMs' output**: Based on the provided documents, when an influenza virus enters a cell, it immediately starts to replicate its genetic material and synthesize viral proteins. Document 1 mentions that virus entering cells initiates specific virus polypeptide synthesis after neutralization of remaining extracellular virus and removal of inhibitors. Document 2 also supports this by explaining that following receptor binding, influenza viruses are internalized and the viral genome is released into the cytosol, followed by further transport to the nucleus where genome replication occurs..
**Prediction**: C (✓) |

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

Table 7: Hyperparameters for retrieval encoder

| Hyperparameter | Value |
|---|---|
| Base model | E5-large-unsupervised |
| Embedding dim | 1024 |
| Embedding pooling | Average at last layer |
| Negative documents | Hard-negatives + in-batch |
| Number of hard-negatives | 5 |
| Softmax Temperature | 0.01 |
| Optimizer | AdamW |
| Learning rate | 2e-5 |
| Batch-size per GPU | 16 |
| Gradient accumulation steps | 2 |
| LoRA Rank | 16 |
| LoRA Alpha | 32 |
| Epochs | 10 |

Table 8: Hyperparameters for critic models

| Hyperparameter | Value |
|---|---|
| Base model | Llama-3.1-8B-Instruct |
| Optimizer | AdamW |
| Learning rate | 1e-5 |
| Batch-size per GPU | 4 |
| Gradient accumulation steps | 2 |
| LoRA Rank | 16 |
| LoRA Alpha | 32 |
| Deepspeed stage | 2 |
| Epochs | 10 |

Dawei Zhu, Nan Yang, Liang Wang, Yifan Song, Wenhao Wu, Furu Wei, and Sujian Li. PoSE: Efficient context window extension of LLMs via positional skip-wise training. In *International Conference on Learning Representations (ICLR)*, 2024.

## A  TRAINING HYPERPARAMETERS

We use 8xRTX A6000 Ada gen 2 for our works. Our implementation is based on the Hugging Face library[2] includes transformers, accelerate, and PEFT libraries. In Table 7 we show our configuration used in retrieval encoder fine-tuning. Table 8 show the settings to train our reward models.

## B  PROMPT FORMATS

### B.1  FEEDBACK COLLECTION

```
System:
You are a search quality rater evaluating the relevance of web pages.
Given a query, a list of correct answers from experts, and a passage
cut randomly from a web page, you must analyze the relevance between the
query and web pages.  you must provide a score on an integer scale of 0
to 2 with the following meanings:
** 2 = highly relevant, provide the correct answer similar to experts
with explanations, very helpful for this query.
** 1 = relevant, provide related information to query but can not find
the correct answer
** 0 = not relevant, should never be shown for this query
```

---

[2]https://huggingface.co

```
Instructions
Split this problem into steps
** Understand the web page
- List all information can be extracted from the web page.
- Only consider information which was clearly mentioned.
- Do not imply or infer based on addition information in your knowledge.
- Do not manipulate.
** Understand the query
- Identify the main subject and intent of the query
- Focusing on specific details like "first," "most," or other qualifiers
that define the query's focus.
** Consider different aspects:
- (match) Does the web page provide information related to the query?
(0/1)
- (gt) Consider the list of correct answer, does the web page mention any
correct answer explicitly with evidences?  (0/1)
- (diff) If the web page does not mention explicitly any correct answer
with evidences, does it provide another answer?  (0/1)
- Note:  a close answer to the correct answer is still wrong.
- Avoid subject mismatching:  for example if the query asks about "The
book thief" and the passage discusses about "The thief", it is different.
** Consider the aspects above, and decide on a final score.  Final score
must be an integer value only.

Your tasks
- Analyze webpage and query by step-by-step mentioned above
- From your analysis, make a final decision.
- Output format:  a json contains 5 keys:  "analyze":  summary your anal-
ysis at most 4 sentences, "match":  0/1, "gt":  0/1, "diff":  0/1, "fi-
nalscore":  0/1/2

Human message:
* Passage:  {passage}
* Query:  {query}
* Correct answer:  {answer}
```

## B.2 QUESTION ANSWERING

**Prompts for NQ and Trivia QA**

```
System:  This is a chat between a user and an artificial intelligence as-
sistant.
The assistant gives helpful, detailed, and polite answers to the user's
questions.
The assistant is provided with 5 passages from Wikipedia.
If there isn't any related information in these passages, answer based on
your knowledge.

Human message:
* Passage 1:  {passage 1}
* Passage 2:  {passage 2}
* Passage 3:  {passage 3}
* Passage 4:  {passage 4}
* Passage 5:  {passage 5}

Query:  {query}
Answer the query directly with the shortest phrase without explanation.
If there are many correct answers, only output one of them.
```

**Prompts for FEVER**

```
System:  This is a chat between a user and an artificial intelligence as-
sistant.
The assistant gives helpful, detailed, and polite answers to the user's
questions.
The assistant is provided with 5 passages from Wikipedia.
If there isn't any related information in these passages, answer based on
your knowledge.

Human message:
* Passage 1:  {passage 1}
* Passage 2:  {passage 2}
* Passage 3:  {passage 3}
* Passage 4:  {passage 4}
* Passage 5:  {passage 5}

Query:  {query}
Answer directly the above query with True or False.
```

