# OpenReview forum: "Reward-RAG: Enhancing RAG with Reward Driven Supervision"
_ICLR.cc/2025/Conference — ICLR 2025 Conference Withdrawn Submission_

### Official Review · Reviewer_gGxB · 2024-11-03

**Soundness:** 2
**Presentation:** 2
**Contribution:** 2
**Rating:** 3
**Confidence:** 3

**Summary:**

This paper introduces building a synthetic dataset approach to train retrieval models leveraging LLM like GPT-4. It first trains a separate reward model estimating relevance of each passage for a given query. To this end, the authors annotate the relevance labels for the synthetic dataset using GPT-4. Then, the constructed dataset is used to fine-tune the retrieval models. The experiments show that efficacy of the synthetic dataset on the general and domain-specific retrieval/QA tasks.

**Strengths:**

Unfortunately, I can not find any strengths in the paper.

**Weaknesses:**

* Overall, the proposed approach does not have technical novelty.
  - First, their claim about RLHF-like alignment framework is entirely unconvincing. They merely trained a reward model and then generated a synthetic dataset for further training. This approach is a common technique employed in numerous studies.
  - Also, it just boosts the performance of the retriever. And there are no specific considerations to the "RAG" aligned with human preference.
  - The achieved performances hardly rely on the proprietary model like GPT-4. They leverage GPT-4 not only as a critical model but also as a generator model in RAG framework. It hinders fair comparison with other baseline models that did not use the GPT-4 model as a generator.
  * It may confuse readers. For example, in Table 2, when their approach is combined with the GPT-3.5-turbo generator, the performances of TriviaQA and FEVER drop significantly.
  * Also, Table 3 does not include the baseline without RAG for the GPT-3.5 turbo (they only report GPT-3.5).

* (As I mentioned above) The experimental results overlook important factors like parameter size, training dataset, computational cost, etc. (Tables 1, 2, and 3).

**Questions:**

* Why do you use the term "CriticGPT"? Is the term general in the research community?
  * I think the critic model could be established with another proprietary model like Claude or Gemini and even with open-source models like LLaMA-70B.
* Please include the parameter size and other useful information to enhance the clarity of the experiments in Tables 1, 2, and 3.
* Why does the fine-tuned retriever even worsen the performances with the GPT-3.5-turbo in Table 2? Does it mean that the synthetic datasets have critical quality issues?

---

### Official Review · Reviewer_Qc54 · 2024-11-03

**Soundness:** 1
**Presentation:** 2
**Contribution:** 2
**Rating:** 1
**Confidence:** 4

**Summary:**

The paper proposes to use RLHF to align RAG encoder to human preference. It leverages the CriticGPT to annotate positive and negative examples to replace actually collecting human feedback. The paper says the reward model it trains "evaluate the relevance between a query and a document", and uses this reward model to finetune the retrieval model. Experiment shows improved retriever performance on NDCG@10. It also shows improved RAG performance based on EM on RAG generation tasks..

**Strengths:**

* Clearly written the contribution and method of the paper.

* innovative idea of using RLHF for RAG encoder model training.

* RAG has been bottlenecked by retrieval quality, and innovation on this front is badly needed and is critical.

* Extensive choice of strong baselines.

**Weaknesses:**

* If the innovation of the paper is in encoder training, authors should devote more experiments to illustrate the improvements of doing RLHF finetuning versus other methods of finetuning. The main Encoder performance results in Table-1 shows results of other encoder models, but essentially no baseline that's comparable to your proposed "E5-large-unsupervised (ours)" which i believe is the RLHF finetuned version. I am not aware of the common datasets used to continue finetune retrievers, but having such a baseline will be helpful to  show the improvement from your method.

* The same problem with Table-2. There's essentially no other baseline that is also built on "E5-large-unsupervised " and then compare RAG performance using the baseline versus your proposed method. I would encourage including a RAG baseline using another retriever that's finetuned on top of "E5-large-unsupervised" using common retrieval datasets.

* The paper clearly needs more polishing. For example, the Table-1's first column should be Encoder models, not "Task", and the captions describing the evaluation benchmarks is confusing.

* RLHF is means aligning to human feedback and preference, but the authors are in fact using a reward model for relevance. So, it's not very accurate, and I recommend changing it sth like contrasting feedback.

**Questions:**

Can you explain and add experiment to illustrate if and how aligning encoder with a relevance reward model improves over other approaches, such as finetune an encoder like E5-large-unsupervised? The current experiment lacks this critical comparison.

---

### Official Review · Reviewer_K5cA · 2024-11-03

**Soundness:** 3
**Presentation:** 4
**Contribution:** 2
**Rating:** 5
**Confidence:** 4

**Summary:**

The paper proposed using reinforcement learning to align RAG with human preferences to achieve better in-domain performance and generalization. To achieve this goal, the authors first use GPT to mimic human annotators and collect a reward model training set, and then use the trained reward model (by MSE) to annotate retrieval results and use InfoNCE to train the retriever. Experiments show the proposed approach could achieve a higher score in both in-domain and out-of-domain tasks.

**Strengths:**

* Clear representation. The authors include all the necessary information in the method section, such as detailed formula and clear figures.

* The method is sound, and the in-domain experiment results are good: the proposed approach beats many baselines despite a smaller retriever.

**Weaknesses:**

* Authors claim the two benefits of the approach: in-domain and out-of-domain (line 161). It is hard for me to understand why the proposed approach can benefit OOD scenarios. The reward model is not trained on the OOD data, so it is not clear how well it can generalize, and the retriever is not trained on the OOD data either. Could authors explain more?

* Therefore I am not surprised that the OOD experiment is not good. The proposed approach performs worse than baselines in all tasks if the inference model is the same (i.e., GPT-4-0613; otherwise, the comparison is not fair). Besides, the authors should include more results of the proposed approach on different inference models to make the table more comparable.

* Table 1 does not include OOD performance, so it is not clear how well the retriever performs on OOD tasks.

Overall, I cannot be convinced that the method benefits OOD scenarios, although the in-domain claim looks sound and is supported by the experiments. Based on the current manuscripts, I will give a weak rejection.

Beyond the above main concern:

* It is suggested to have an analysis of the reward model: how well it aligns with humans (GPT) and how well it generalizes.

* Though the current approach is sound, it does not bring much new insight and novelty.

**Questions:**

* In Table 2, what is the LLM in the retrieved RAG baseline?  The table only lists the retriever.

* In Table 3, can the authors give more details of the RRF-4 methods? My understanding is that RRF-4 is a re-ranking algorithms, so what are the retrievers used in Table 3?

* Does the Emb_p contains the information of query (i.e., if the [SEP] token will separate the attention)?


Type:

Line 360: Table 4.2 -> Table 3

---

### Official Review · Reviewer_ou62 · 2024-11-04

**Soundness:** 2
**Presentation:** 2
**Contribution:** 2
**Rating:** 3
**Confidence:** 5

**Summary:**

This paper introduces a method called RewardRAG that integrates reward modeling into RAG systems. The main contribution is to introduce a new method to generate synthetic data for data augmentation to train an improved retrieval model. The way they create synthetic data is somewhat new, but due to the limited experimental setup, it is not clear if the proposed method is helpful or not.

**Strengths:**

The proposed method for synthetic data generation is new.

**Weaknesses:**

To better understand the impact of the generated synthetic data, one needs to do a more rigorous evaluation for retrieval. For example, why their proposed method works better than re-ranking? In table 1, they mostly compare with older and smaller encoder models, and even in that case, their method only wins on the NQ dataset.

In table2, the base LM for RewardRAG is GPT40/chatgpt, and it is not clear how the RewardRAG model compares with GPT4 with the basic RAG model. Since RewardRAG's contributions are on improving retrieval, the authors should compare with the same underlying LM and varying retrievers e.g., GPT4o with their retrievers v.s. other retrievers (edited)
In Table 2, they compare RewardRAG with other models with different base LMs, for example Self-RAG is trained on a 7B and 13B models, and we know that GPT4o is significantly better than most 13B scale models.

Base on Table 3, their number is much worse that GPT4o+standard RAG? (they only outperform GPT4o+RAG on PubmedQA)

**Questions:**

Please check weaknesses above.

---

### Note · Authors · 2024-11-19

I have read and agree with the venue's withdrawal policy on behalf of myself and my co-authors.